# Structure, Functionality, Compatibility with Pesticides and Beneficial Microbes, and Potential Applications of a New Delivery System Based on Ink-Jet Technology

**DOI:** 10.3390/s23063053

**Published:** 2023-03-12

**Authors:** Mohamed Idbella, Domenico Giusti, Gianluca Gulli, Giuliano Bonanomi

**Affiliations:** 1Department of Agricultural Sciences, University of Naples Federico II, Via Università 100, 80055 Portici, Italy; 2Laboratory of Biosciences, Faculty of Sciences and Techniques, Hassan II University, Casablanca 28806, Morocco; 3STMicroelectronics, Via C. Olivetti 2 Agrate Brianza (MB), 20864 Agrate Brianza, Italy; 4Task Force on Microbiome Studies, University of Naples Federico II, 80138 Naples, Italy

**Keywords:** sustainable farming, beneficial microbes, droplet size, fungicides, herbicides, microgreens

## Abstract

Accurate application of agrochemicals is an important way to achieve efficient use of chemicals and to combine limited pollution with effective control of weeds, pests, and diseases. In this context, we investigate the potential application of a new delivery system based on ink-jet technology. First, we describe the structure and functionality of ink-jet technology for agrochemical delivery. We then evaluate the compatibility of ink-jet technology with a range of pesticides (four herbicides, eight fungicides, and eight insecticides) and beneficial microbes, including fungi and bacteria. Finally, we investigated the feasibility of using ink-jet technology in a microgreens production system. The ink-jet technology was compatible with herbicides, fungicides, insecticides, and beneficial microbes that remained functional after passing through the system. In addition, ink-jet technology demonstrated higher area performance compared to standard nozzles under laboratory conditions. Finally, the application of ink-jet technology to microgreens, which are characterized by small plants, was successful and opened the possibility of full automation of the pesticide application system. The ink-jet system proved to be compatible with the main classes of agrochemicals and showed significant potential for application in protected cropping systems.

## 1. Introduction

According to the current scenario, the human population will reach 9.1 billion in 2050, and providing sufficient food is a major challenge for agriculture in the future. In this context, careful application of agrochemicals is an important way to achieve efficient use of chemicals and combine limited environmental pollution with effective control of weeds, pests, and diseases that still cause significant yield losses worldwide. In conventional, as well as organic farming systems, weeds, pests, and diseases are largely controlled with synthetic or natural products applied to crops with various sprays [1]. Smart and precision farming based on robotics, machine automation, location technology, and advanced data analysis including the Internet of Things [2,3], have been widely studied to increase the efficiency of fertilizers and pesticide use to maximize crop yields and reduce production losses due to pest and diseases [4,5].

Under field and greenhouse conditions, boom spraying is the most used technique. The agrochemicals are usually diluted in water and then distributed on the plant by atomizing the liquid into droplets of different sizes through nozzles or atomizers in the air stream. In classic boom systems, the pressure of the sprayer provides the initial velocity of the droplets, which, in conjunction with gravity, transport the products to the target area of the plant. In addition, some sprayers use an intense airflow generated by either hydraulic or pneumatic machinery to assist in the transport of droplets from the nozzle to the target plant [6].

When applied in the field, partial loss of agrochemicals to the environment is inevitable as droplets drift away from the target crop or volatilize during application [7]. To optimize agrochemical application, the farmer must regularly maintain the spray system and choose the correct sprayer settings in terms of nozzle type and droplet size [8]. Choosing the right droplet size is a crucial step because small droplets can easily drift during transport from the nozzle to the target crop, especially if their diameter is less than 100 μm [9]. On the other hand, large droplets with a diameter of >500 μm are not drifting and do not volatilize because they reach the target plant in a few seconds but do not provide continuous surface coverage, which is a major problem for fungicides and insecticides. Selecting the appropriate spray technology to optimize droplet diameters that maximize crop coverage and limit drift is a challenge for farmers working under variable environmental conditions resulting from crop type, boom height, tractor speed, and microclimatic conditions, especially wind speed [10]. In recent decades, numerous studies have been conducted to improve the efficiency of agrochemical applications, including continuous development of sprayer components such as the lance and nozzle [11].

Despite continuous development of sprayer components, no breakthrough technology has emerged in this field in recent years. Here we introduce a new spraying technology for agricultural applications based on the ink-jet system. Overall, ink-jet applications are an important area of microelectromechanical systems. The general principle underlying ink-jet technology is the controlled ejection of a liquid, whereby a material dispersed or dissolved in the liquid is ejected in a very precise and controlled manner. Drop-on-demand is a newer technology whose main players are thermal ink-jet and piezoelectric jet. The main feature is the ability to spray drops of liquid on demand. Both technologies have their own strengths and weaknesses, and depending on the fluid, one may be preferable to the other. Historically, the thermal ink-jet process was invented and developed simultaneously by Canon and Hewlett Packard in the late 1970s and early 1980s [12]. This process is still used in a large proportion of ink-jet printers for home and office use. Briefly, there is a small heating resistor in a cavity behind the nozzle that is in good thermal contact with the fluid. Rapid heating within a few microseconds results in the overheating and vaporization of a thin layer of liquid in contact with the heating element: a vapor bubble rapidly expands, creating the liquid displacement and energy necessary to force a droplet out of the nozzle. Once the droplet is expelled, the heat source is turned off and the bubble quickly collapses, drawing in fresh liquid for the next pulse. The second pulse starts again to create another drop. A key advantage of this technique is that the actuator is simply a resistive track that can be fabricated using multilayer manufacturing processes such as standard microlithography techniques to ensure precision, repeatability, and quality of the product in mass production. The fluid formulation plays a key role for thermal ink-jet, as it should be able to withstand heating and evaporation processes and limit the deposition of components on the heating surface or in the nozzle (kogation). In addition, a certain amount of water or other easy-to-boil liquid is needed to promote evaporation. Recently, ink-jet printing emerged as the forefront for the biosensor manufacturing approach, including point-of-care diagnostic biosensors [13,14].

In this context, the general objective of this work is to investigate the potential application of a new delivery system based on ink-jet technology for agricultural applications. To achieve this goal, we first evaluated the compatibility of ink-jet technology with a range of herbicides, fungicides, insecticides, and beneficial microbes. This step was necessary because ink-jet technology relies on heating liquids to high temperatures, which could alter the functions and biological activity of agrochemicals. In a second step, we evaluated the potential of ink-jet technology for agrochemical reduction by investigating droplet size optimization for herbicides and insecticides that, globally, are economically important crop protection products. Finally, we evaluated the potential application of ink-jet technology in a real-world cropping system. Given the small size of current ink-jet technology and the limited volumes that can be sprayed in a unit of time, it was decided to apply the technology to microgreens production systems [15]. Microgreens are fresh-cut plants intended for fresh consumption, and their market has expanded rapidly in recent years [16]. As the name microgreens suggests, they are vegetable crops with a very short cycle, often less than 30 days, and with small dimensions that can be grown even in artificial, highly automated systems [17]. All these characteristics made microgreens the ideal system for the first attempts to apply ink-jet technology to real production systems. To this end, we developed a dedicated system, DEMO, that combines ink-jet technology with an automatable product application system. Specific aims of the study were:i.Testing the compatibility of the new system with herbicides, fungicides, insecticides, and beneficial microbes as well.ii.Assessing the potentiality of the new system for reduction of chemicals by optimization of droplet size.iii.Provide a test for potential application to a microgreens production system.

## 2. Materials and Methods

### 2.1. Description of the New Sprayer System

Thermal ink-jet designs can have different configurations based on the position of the heating element relative to the nozzle (Appendix A). The most common configurations are the front shooter (also called roof shooter) and the side shooter. Thermal ink-jet technology is based on rapid and sudden boiling of the fluid in contact with the heater. This creates a pressure peak in the combustion chamber, which results in the liquid exiting the nozzle. The principle of operation is based on the process of bubble growth. When a heat transfer surface is in contact with a liquid, bubble formation can generally occur by spontaneous nucleation, either heterogeneous or homogeneous. The former refers to bubble formation due to nucleation sites, gas pits, or defects on the heating surface, while in the latter case, heat disturbances in the fluid itself are the driving force. Thermal ink-jet technology uses very rapid heating on the order of 100 °C/μs. Under these conditions, the liquid becomes a superheated fluid well above its normal boiling point, and explosive and rapid vaporization is known to occur. Most agrochemicals are commonly diluted in water due to the high heat transfer capacity and detonation of water. The boiling temperature of water is 100 °C at 1 atm, but when heated rapidly, the boiling temperature of water increases to nearly over 300 °C. The bubbles formed under these conditions are the driving force for the liquid jet. If the driving conditions are suitable for the heater configuration and the fluid selected, the surface is immediately covered by many small nuclei that combine to form a thin film of vapor before they become large. The bubbles thus formed grow by inertia and increase the internal pressure until equilibrium with the external ambient pressure is reached and the bubbles shrink. Only when a certain minimum amount of energy, called turn-on energy, is provided by the drive signal, do the bubbles grow and the generated pressure can expel the liquid. The overall process is summarized in Figure 1, which shows the evolution of the bubbles with time. The onset of boiling is around 250 °C and corresponds to the surface covered by many small bubbles. The driving conditions provide sufficient energy, and the bubbles grow and fuse, creating an optimum jet pressure. If too little energy is supplied to the heater, ineffective bubble growth will occur. Since the bubbles are generated from a superheated liquid thin film, the size of the bubbles depends on the temperature of the liquid before the boiling starts, which is reflected in the droplets formed. The temperature of the ejected droplets is at room temperature, since only 1% of the total liquid present in the chamber is involved in bubble formation.

The system design consists of (i) a daughter board named Printed Circuit Board (PCB); (ii) a main driver control board. The daughter board PCB should look like Figure 2, with a die on one side of the PCB, aligned with a fluid passage; on the opposite side of the PCB, a luer connector to connect the die to a fluid supply tube. The daughter board is then glued to a plastic cap and then clamped to the mechanical characterization device. The master driver controller board is mainly an extension of the STMicroelectronics STEVAL-STLKT01V1 SensorTile Development Kit with adequate functions installed to control the individual heating elements on the die. The SensorTile is a tiny square IoT module with powerful processing capabilities. The module features an 80 MHz STM32L476 microcontroller with ultra-low power consumption and Bluetooth Low Energy connectivity based on the BlueNRG network processor. The SensorTile also features a wide range of motion and environmental sensors, MEMS, including a microphone. The STEVAL-STLKT01V1 development kit comes with a set of cradle boards that enable hardware scaling, as well as software/firmware libraries and tools. With the app ready to use, SensorTile is a true IoT design lab (https://www.st.com/en/evaluation-tools/steval-stlkt01v1.html, accessed on 15 July 2021).

### 2.2. Compatibility Test of Thermal Ink-Jet with Alive Beneficial Microbes

The temperature of the ejected droplets at thermal ink-jet is at room temperature because only ~1% of the total liquid present in the chamber is involved in bubble formation. These theoretical considerations suggest that the use of this technology would be compatible with agrochemicals, since only a small fraction of the liquid is exposed to high temperature and for a very short time (on the order of milliseconds). However, we have experimentally investigated the compatibility of thermal ink-jet with several agrochemicals. Specifically, five microorganisms were used, namely *Bacillus subtilis*, *Coniothyrium minitans*, *Glomus mosseae*, *Pseudomonas fluorescens*, and *Trichoderma harzianum*, which are of interest to agriculture because they are commonly used as biostimulants and biological control agents (Appendix A). The strains of the species used were available in the laboratory of the Department of Agriculture. Viability tests were performed using different methods based on the biology of the organism in question: For example, mycorrhizal fungi such as *G. mosseae* do not grow on substrates under laboratory conditions. The methods used for each microorganism are listed below.

#### 2.2.1. *Trichoderma harzianum*

The tests were performed by two methods: Spore germination and plate growth on agar as solid substrate [18]. The fungus was kept on solid substrate PDA (Potato Dextrose Agar—Oxoid). Spores, also called conidia, were obtained by adding 10 mL of sterile water to Petri dishes (Ø 90 mm) containing the fungal cultures. The cultures were scraped on the surface with a steel rod to facilitate detachment of the conidia from the fungal mycelium. A spore suspension was collected and then filtered through a glass wool filter, centrifuged (2395× *g* for 5 min), rinsed three times with sterile water, and diluted to a concentration of 1 × 10^6^ conidia mL^−1^ by reading on a hemocytometer. Then, the conidia were ejected using the ink-jet device with the following setting parameters: P = pulse width in (nano second): 1500 (ns); V = Voltage: 15 (V); F = frequency (Hz): 2800 (Hz). The spores were collected in a 15 mL Falcon tube after the ink-jet spray treatment. The resulting spores were used for both the germination test and plate growth. For the germination test, a suspension of PDB (Potato Dextrose Broth—Oxoid) with 5 mM phosphate buffer at pH 6.7 was used. Then, 5 μL of the conidia suspension at a concentration of 1 × 10^6^ mL^−1^ was added to 50 μL of the PDB suspension in a 96-well ELISA plate (FALCON) and incubated at 25 °C. The number of germinated spores was determined by optical reading under a microscope after 36 h of incubation. Spores obtained by the same procedure but not sprayed with the ink-jet device served as controls. Germination tests were performed with 10 experimental replicates and data were statistically analyzed using analysis of variance (one-way ANOVA) followed by Duncan’s test. For the test of spore production on plates, 200 μL of the conidial suspension at a concentration of 1 × 10^6^ mL^−1^ was added to a Petri dish containing PDA and then spread with a steel rod. The plates were coated with Parafilm and incubated at 25 °C for 7 days. After 7 days, conidia production was quantified by adding 10 mL of sterile water, scraping the surface of the culture with a steel rod to detach the conidia, and then counting with a hemocytometer. Additionally, in this case, spores obtained by the same procedure but not sprayed with the ink-jet device were used as controls. Data on conidia production were performed with 10 experimental replicates and statistically analyzed using analysis of variance (one-way ANOVA) followed by Duncan’s test.

#### 2.2.2. *Coniothyrium minitans*

Conidia of *C. minitans* were produced using the same methods as described for *T. harzianum*. ELISA plates were incubated at 20 °C and spore germination was determined after 48 h. As for the assay of spore production in plates, the test was performed using the same methods described for *T. harzianum*, with the only difference that the incubation period was 21 days. In both tests, untreated spores with the ink-jet device were used as controls, and data on germination and conidia production were statistically analyzed by analysis of variance (one-way ANOVA) followed by Duncan’s test.

#### 2.2.3. *Glomus mosseae*

*G. mosseae* is an obligate symbiotic fungus and therefore relies on mycorrhizal association with plant roots to complete its life cycle. *G. mosseae* cannot be grown in the laboratory without a host plant, so the only possible test to evaluate its viability is spore germination. *G. mosseae* was cultivated in sorghum (*Sorghum vulgare*) grown in pots (Ø 24 cm) in the greenhouses of the Department of Agriculture and renewed every two months. Spores of *G. mosseae* were obtained by wet sieving from the soil on which sorghum was grown. For this purpose, 250 mL of moist soil was suspended in 1 L of water and stirred for 30 min. The soil suspension was sieved through a series of sieves with progressively smaller diameters from 1 mm to 40 μm. The material collected from the 40-μm sieve was centrifuged (2395× *g* for 5 min), and the pettel was suspended in sterile water on a slide. Spores were collected by hand with a surgical needle and then rinsed five times in sterile water. The spores were stored in a refrigerator at +4 °C. The following growth medium was used for the germination assay: KCl, 4.0 mg; KNO_3_, 6.4 mg; MgSO_4_ 7H_2_O, 4.0 mg; Ca(H_2_PO_4_) 2H_2_O, 0.8 mg; FeNaEDTA, 0.19 mg; thiamine, 0.4 mg; biotin, 0.04 mg; sucrose, 1.0 g; Bacto-Agar (Difco Mich.), 10 g. The medium was adjusted to pH 6.6 and autoclaved. Ten mL of the culture medium was applied to Petri dishes and allowed to solidify. The spore suspension of *G. mosseae* was treated with the ink-jet system, and the spores were then spread on the substrate described above. Germination was monitored every 2 days for a total of 14 days. Spores that were not treated with the ink-jet system served as controls. Germination data were statistically analyzed using analysis of variance (one-way ANOVA) followed by Duncan’s test.

#### 2.2.4. *Bacillus subtilis*

*B. subtilis* was grown in vitro at 37 °C. The following substrate was used as culture medium: 1% casein hydrolysate, 0.47% L-glutamate, 0.16% L-asparagine, 0.12% L-alanine, 1 mM KH_2_PO_4_, 25 mM NH_4_Cl, 0.22 mg mL^−1^ Na_2_SO_4_, 0.2 mg mL^−1^ NH_4_NO_3_, 1 g mL^−1^ FeCl_3_-6H_2_O, 25 mg L^−1^ CaCl_2_-2H_2_O, 50 mg L^−1^ MgSO_4_, 15 mg L^−1^ MnSO_4_-H_2_O, 20 *g* mL^−1^ L-tryptophan, adjusted to pH 7.0 and sterilized by autoclaving. The viability of *B. subtilis* cells after treatment with the ink-jet system was evaluated in liquid culture by spectrophotometric measurement (Beckman DU-520, Beckman Coulter, Inc., CA, USA) of optical density at 600 nm (OD600). Specifically, 100 μL of the suspension of *B. subtilis* cells treated with the ink-jet system was added to 10 mL of the culture substrate described above. The material was incubated at 37 °C and the optical density was measured after 6, 12, and 24 h of incubation. The optical measurement data were statistically analyzed by two-way analysis of variance (ANOVA) using incubation time and cell treatment as analysis factors.

#### 2.2.5. *Pseudomonas fluorescens*

*P. fluorescens* was grown in vitro at 28 °C according to the methods already described for *B. subtilis*, but using LB (Lysogeny Broth, Fluka, Sigma-Aldrich, MO, USA) as substrate. The viability assay was also performed in the same manner by inoculating 100 μL of the suspension of *P. fluorescens* cells treated with the ink-jet system into 10 mL of the LB culture substrate. The material was incubated at 28 °C and the optical density was measured after 6, 12, and 24 h of incubation. The optical measurement data were statistically analyzed using two-way analysis of variance (ANOVA), using incubation time and cell treatment as analysis factors. In addition, for demonstration purposes, the cell suspension of *P. fluorescens* was directly applied to a Petri dish containing LB substrate with the addition of agar and then solidified using the ink-jet system.

### 2.3. Compatibility Test of TIJ with Herbicides

The objective of the tests was to evaluate the efficacy of herbicides when using the ink-jet technology. Herbicides are plant protection products used to eliminate weeds that compete with crops and interfere with production. Two herbicides were tested, glyphosate and cicloxidim at five concentrations, with two weed species: *Lolium temulentum* L. and *Avena fatua* L. (Appendix A). *L. temulentum* is an annual weed species belonging to the grass family. *A. fatua* is also an annual weed belonging to the grass family and is considered one of the most important weeds for many agricultural crops. Glyphosate is the most widely used crop protection product in the world [19]. Cycloxidim is a systemic herbicide that targets grasses post-emergence and is therefore used when weeds have already grown out of the soil. Cycloxidim is taken up by the leaves but is transmitted systemically and therefore is also active in the underground organs of the plants. The herbicidal effect of both glyphosate and cycloxidim occurs within 5–7 days of product application. For the biological tests, the experimental protocol included the following phases: Seeding of *L. temulentum* or *A. fatua* in Magenta GA7 containers for vegetable crops containing vermiculite and previously sterilized in an autoclave. For seeding, 30 seeds per magenta were used for *L. temulentum* and 15 seeds per magenta for *A. fatua*. The containers were incubated in a growth chamber with a day/night cycle of 14/10 h at a temperature of 22 ± 4 °C. After 10 days of incubation, when the seeds had germinated and reached a height of about 4 cm, the herbicides were applied. The application with an ink-jet device was carried out with ‘die’ that produced droplets with a diameter of 20 μm and 50 μm, hereafter referred to by the abbreviations ‘ST20’ and ‘ST50’. Treatments with the ink-jet device were compared to a control that was not treated with herbicides and to a treatment that was applied with a manual pressure spray system. This system was used because of its ease of use at small volumes (<10 mL), useful under laboratory conditions. This treatment will be referred as ‘spray’. During the experiments, the volume of sprayed solution was kept constant for each treatment. Specifically, 200 μL of solution was applied for each experimental unit (magenta container with 20 *Lolium* plants). This volume was determined by calculating the volume of the filling tube of the ink-jet system as well as the spray system. The volume used was sufficient to completely wet the leaf surface of the plants tested. The two herbicides, glyphosate and cycloxidim, were applied at five concentrations (Appendix A). The highest concentration was determined based on the maximum recommended dose on the label (Clinic 360 for glyphosate and Stratos for cycloxidim). Subsequent concentrations were performed after serial dilution with a logarithmic factor. Seven days after the application of the herbicides, the vegetative state of the plants was evaluated by visual and photographic surveys. Images were then analyzed using ImageJ software to quantify the percentage of yellowed leaf surface, a parameter used as an index of herbicide treatment efficacy. A total of four trials were conducted: 1—glyphosate on *L. temulentum*; 2—glyphosate on *A. fatua*; 3—cycloxidim on *L. temulentum*; 4—cycloxidim on *A. fatua*. Each experiment included 16 treatments: untreated control, spray herbicide at five strengths, ST20 at five concentrations, and ST50 at five concentrations. Each treatment was replicated three times, resulting in a total of 48 experimental units. The data obtained in each trial, expressed as percentage of yellowed leaf area, were analyzed after logarithmic transformation using two-way analysis of variance (ANOVA), with herbicide concentration and application system as variables.

### 2.4. Test for Reducing Application Rate of Insecticides

The objective of the tests was to evaluate the efficacy of insecticides when applied with ink-jet technology. Insecticides are crop protection products used to control insects, one of the most damaging groups of arthropods in agriculture. Worldwide, insecticides are used second only to herbicides [20,21]. Two insecticides, namely abamectin and deltamethrin, were tested at five concentrations with two insect species: *Galleria mellonella* L. and *Sarcophaga carnaria* L. (Appendix A). *G. mellonella* and *S. carnaria* are not phytophagous species and are of limited agricultural interest. Therefore, the choice was dictated by two reasons: (i) They are two species that are easy to find commercially and easy to grow; (ii) They represent two important insect orders, Lepidoptera for *G. mellonella* and Diptera for *S. carnaria*.

Abamectin is an active ingredient based on avermectin B1, a compound derived from the actinobacterium *Streptomyces avermitilis*. By mechanism of action, abamectin is neurotoxic and interferes with nerve and muscle transmission, resulting in paralysis of the insects that ingest it and secondarily come into contact with it. Deltamethrin is an insecticide that acts primarily by contact on insects. Deltamethrin has a neurotoxic effect and has a strong knockout behavior that immobilizes insects. It has a rapid onset of action, as early as 48 h, and a broad spectrum of activity. For the biological tests, the experimental protocol included the following phases: Positioning of *G. mellonella* larvae in Petri dishes with a diameter of 3.3 cm and of *S. carnaria* in plastic containers with six wells with a diameter of 2 cm. Specifically, three *G. mellonella* larvae per plate and 10 larvae of *S. carnaria* per well were used (Appendix A). The difference in number is due to the size of the larvae, which is larger in the case of *G. mellonella*. The containers were incubated in a growth chamber with a day/night cycle of 14/10 h at a temperature of 22 ± 4 °C. Insecticides were applied after 2 days of adaptation. Application with an ink-jet device was performed with two settings of the ink-jet system that produced droplets with diameters of 20 μm and 50 μm, referred to by the acronyms ST20 and ST50. Treatments with the ink-jet system were compared to a control not treated with insecticides and to a treatment with a manual pressure spray system. During the tests, the volume of solution sprayed was kept constant for each treatment. In detail, 200 μL of solution was applied for each experimental unit. This volume was determined by calculating the volume of the filling tube of the ink-jet system and the spray system. The volume used was sufficient to completely wet the body of the larvae subjected to biological tests. The two insecticides, abamectin and deltamethrin, were used at five concentrations. The highest concentration was determined based on the maximum dose indicated on the label (Vertimec for abamectin and Decis for deltamethrin). Subsequent concentrations were determined by serial dilution with a logarithmic factor (Appendix A). Seventy-two hours after insecticide application, the vital status of larvae was quantified by assessing their mobility. The percentage of dead larvae was used as a parameter for the efficacy of the insecticide treatments. A total of four trials were conducted: 1—abamectin on *G. mellonella*; 2—deltamethrin on *S. carnaria*; 3—deltamethrin on *G. mellonella*; 4—deltamethrin on *S. carnaria*. Each experiment included 16 treatments: untreated control, spray herbicide at five strengths, ST20 at five concentrations, and ST50 at five concentrations. Each treatment was replicated five times, resulting in a total of 80 experimental units. A total of 240 larvae were used for each *G. mellonella* experiment, while 800 larvae were used for *S. carnaria*. The data obtained in each experiment, expressed as percentage of dead larvae, were analyzed after logarithmic transformation by two-way analysis of variance (ANOVA), using the concentration of the insecticide and the application system as variables.

### 2.5. Test for Reducing Application Rate of Beneficial Microbes by Modulating Droplet Size

The aim of the tests was to evaluate the effectiveness of beneficial microorganisms when ink-jet technology is applied. Beneficial microorganisms are living organisms used in both conventional and organic agriculture to promote plant growth and control plant diseases [22,23,24]. Globally, the use of beneficial microorganisms has steadily increased in recent decades [25,26]. Two beneficial microbes, namely *T. harzianum* and *B. subtilis*, were tested at five concentrations with two plant species of agricultural interest: *Lolium perenne* L. and *Eruca vesicaria* L. (Appendix A). *L. perenne* is a species belonging to the grass family that is commonly used as a mowing plant in meadows and pastures. This plant is also used to establish lawns in green areas such as public and private parks and soccer fields. *E. vesicaria*, common rocket, is a plant of the cruciferous family and is commonly used as a leafy vegetable. In recent years, cultivation of this species has become widespread, especially in four-range systems and, more recently, as microgreens.

Fungi of the genus *Trichoderma* are marketed in dozens of countries around the world. These fungi are used both in powder form and, more commonly, as a spore suspension at a concentration of 1 × 10^4^ to 1 × 10^8^ per mL. *T. harzianum* is one of the most used species because it not only stimulates plant growth but also can control hundreds of pathogens from the viral, bacterial, and fungal classes [27]. *B. subtilis* is a Gram-positive bacterium that belongs to the phylum Firmicutes. It is widely used as a biocontrol agent, and dozens of strains of *B. subtilis* are currently marketed for their biostimulant activity as well as for the control of numerous plant diseases caused by bacteria (e.g., *Erwinia*, *Xanthomonas*), and fungi (e.g., *Alternaria*, *Rhizoctonia*, *Pythium*, *Fusarium*, *Phytophthora*) [28]. It is marketed in liquid, concentrated liquid, and powder forms. In this study, a liquid product containing a suspension of bacterial cells was used. For the biological tests, the experimental protocol included the following phases: Seeding of *L. perenne* and *E. vesicaria*. *L. perenne* was grown in Magenta GA7 plant culture containers containing vermiculite and previously autoclaved. Thirty seeds per magenta were used. For *E. vesicaria*, square plates (12 × 12 cm) filled with vermiculite and seeded with 50 seeds per plate were used. The magenta containers and square plates were incubated in a growth chamber with a day/night cycle of 14/10 h at a temperature of 22 ± 4 °C. After 7 days of incubation, when the seeds had germinated and reached a height of about 4 cm for *L. perenne* and 2 cm for *E. vesicaria*, the beneficial microorganisms were deployed. The application with the device was performed with the ink-jet setting, which produces droplets with a diameter of 20 μm ST20 and 50 μm ST50. Treatments with an ink-jet device were compared to a control not treated with beneficial microorganisms and to a treatment with a manual pressure spray system. During the tests, the volume of solution sprayed was kept constant for each treatment. Specifically, for each experimental unit, 200 μL of suspension was applied for *L. perenne* and 400 μL for *E. vesicaria*. The volume used was sufficient to completely wet the leaf surface of the tested plants. The two beneficial microorganisms were used at five concentrations, expressed as spore counts for the fungus and cell counts for the bacterium. The highest concentration was determined based on the maximum doses reported in the literature [27]. Subsequent concentrations were performed by serial dilution by a factor of ten (Appendix A). Fifteen days after application of the beneficial microorganisms, the vegetative status of the plants was assessed using destructive biometric measurements. Plants were harvested, roots washed from vermiculite residues, dried in a ventilated oven at 40 °C for 72 h, and then weighed using a precision balance. The total dry weight of the plants, expressed as a percentage of the control not treated with beneficial microorganisms, was used as a parameter to evaluate the effectiveness of the treatments performed. A total of four experiments were conducted: 1—*T. harzianum* on *L. perenne*; 2—*T. harzianum* on *E. vesicaria*; 3—*B. subtilis* on *L. perennial*; 4—*B. subtilis* on *E. vesicaria*. Each experiment included 16 treatments: untreated control, spraying with beneficial microorganisms at five concentrations, ST20 at five concentrations, and ST50 at five concentrations. Each treatment was repeated three times, resulting in a total of 48 experimental units. The data obtained from each trial, expressed as dry weight as a percentage of the control, were analyzed after logarithmic transformation using a two-way analysis of variance (ANOVA), with the concentration of beneficial microorganisms and the application system as variables.

### 2.6. Application of TIJ System to Microgreens

Microgreens are fresh-cut plants intended for fresh consumption and have experienced a strong market expansion in recent years [16]. As the term microgreens implies, they are vegetable crops with a very short cycle, often less than 30 days, and with small dimensions that can be grown in artificial systems with a high degree of automation [17]. All these characteristics made the microgreens an ideal system for the first attempts to apply the ink-jet technology to real production systems. To this end, we developed a dedicated system DEMO that combines ink-jet technology with an automatable product application system. The structure of DEMO consists of a suitably modified 3D printer (model GEEETech I3 Pro B, Shenzhen Getech Technology Co., Ltd., Shenzhen, China) (Appendix A). The DEMO was modified by replacing the extruder with a carrier that can accommodate the ink-jet technology. In the current configuration, the ink-jet technology is still manually controlled using the protocol described above. On the other hand, the horizontal and vertical movement of the platform on which the test organisms rest is controlled by the Repetier-Host V2.2.4 software. With this software, it is possible to program specific trajectories of the platform by acting on the three variables of x, y, and z space. It is also possible to control the locomotion speed of the system. Appendix A shows two examples of programmed ‘paths’ used for application testing. Overall, DEMO proved to be a relatively easy system to program, adaptable to crops less than 20 cm tall, and offers further automation potential with respect to application to micrograss cropping systems.

#### 2.6.1. Drop Distribution Test with Hydrosensitive Papers

A very important aspect that affects the effectiveness of the application of plant protection products is the size of the droplets that are distributed on the plant surface. In general, very small droplets (<30–50 μm in diameter) are highly subject to drift associated with convective air movements. In contrast, very large droplets (>500 μm in diameter), although hardly affected by drift once they reach the surface of the plant, are subject to gravity due to their size and are therefore subject to the phenomenon of dripping. Both drift and dripping are processes that reduce the effectiveness of plant protection products’ application [29]. To optimize the variables involved and evaluate the functionality of spray systems, water-sensitive cards can be used (SpotOn Paper, Innoquest Inc., Woodstock, IL, USA). These 51 × 76 mm papers change color from yellow to blue in the contact area when in contact with liquids. Evaluation of the color changes in relation to the overall coverage and spatial distribution of the droplets makes it possible to evaluate the distribution efficiency of a liquid. Based on these premises, water-sensitive rolling papers were used as the first test to evaluate ink-jet technology at DEMO. In the first test, the ink-jet device was compared to an ink-jet system producing droplets with a diameter of 20 μm (ST20) and 50 μm (ST50). The treatments with the ink-jet device were also compared to an experiment performed with a manual pressure spray system. This system was used because of its ease of use (spray) at small volumes (<10 mL) in laboratory conditions. During the tests, the volume of sprayed solution was kept constant for each treatment. Specifically, 300 μL of the solution was applied for each experimental unit (water-sensitive paper). This volume was determined by calculating the volume of the filling tube of the ink-jet system and the spray system. A constant height of 10 cm from the platform was maintained in all treatments, and the liquid applied was water. Appendix A summarizes the treatments and variables considered in the first experiment. In the second experiment, the ST20, the ST50, and the spray systems were compared under the same experimental conditions described above, changing only the height from the platform. Specifically, four heights were compared: 5, 10, 15, and 20 cm. The results of both experiments were quantified by visual and photographic examinations 5 min after each application. The photographic images were then analyzed using ImageJ software to quantify the percentage of the surface of the card that was still yellow or had turned blue and therefore had been irradiated by the applied liquid.

#### 2.6.2. Test with Beneficial Microbes

The aim of the tests was to evaluate the efficacy of beneficial microbes and, in particular, *T. harzianum*, which had already been successfully used in the experiments described in the previous sections when applied with the ink-jet technology integrated in DEMO. *T. harzianum* was tested with two plant species of interest for agriculture in fresh-cut systems: *E. vesicaria* and *L. sativa*. In recent years, cultivation of these species has become widespread, especially in fresh-cut systems and, more recently, as microgreens.

For the biological tests, the experimental protocol included the following phases: Seeding of *L. sativa* and *E. vesicaria*. Both species were grown in 14 cm diameter plastic pots that previously contained soil sterilized in an autoclave. Sowing was conducted by using 20 seeds per pot. The pots were incubated in a growth chamber with a day/night cycle of 14/10 h at a temperature of 25 ± 4 °C. After 10 days of incubation, when the seeds germinated and the plants were about 4 cm high, the beneficial microorganisms were applied. The application with an ink-jet device was carried out with the ST20 and ST50 systems. The treatments with the ink-jet device were compared with a control that was not treated with beneficial microorganisms and with a treatment that was applied with a manual pressure spray system. During the tests, the volume of solution sprayed was kept constant for each treatment. Specifically, 500 μL of suspension was applied for each experimental unit. *T. harzianum* was used at a concentration of 1 × 10^5^ mL^−1^, which was found to be optimal in the experiments described in the previous sections. The concentration of 1 × 10^5^ mL^−1^ proved to be an ideal compromise, as it is able to stimulate plant growth but does not cause clogging problems of the ST20 system (see Section 3). Fifteen days after application of the beneficial fungus, the vegetative state of the plants was assessed by destructive biometric measurements. Plants were harvested, roots washed of soil debris, dried in a ventilated oven at 40 °C for 72 h, and then weighed using a precision balance. The total dry weight of the plants, expressed as a percentage of the control not treated with beneficial microorganisms, was used as a parameter for evaluating the effectiveness of the treatments performed. In total, the experiment included 10 treatments: untreated control, beneficial microorganism’s spray on 10 cm, 15 cm, and 20 cm, ST20 on 10 cm, 15 cm, and 20 cm, and ST50 on 10 cm, 15 cm, and 20 cm (Appendix A). Each treatment was repeated three times, providing a total of 30 experimental units for each crop. Data obtained from each trial, expressed as dry weight as a percentage of control, were analyzed by analysis of variance (ANOVA).

## 3. Results and Discussion

### 3.1. Compatibility of TIJ with Benefical Bacteria and Fungi

Germination of *T. harzianum* spores was very high (>90%) in both the control and after treatment with the ink-jet spray system, with no significant differences between the two treatments (Figure 3). The production of *T. harzianum* conidia after one week of incubation was comparable between the ink-jet and spray treatments, with no statistically significant differences (Figure 3). Additionally, from a macroscopic point of view, the plates obtained from the spores treated with the ink-jet system did not differ from the control plates and showed the typical green color (Figure 3).

Germination of *C. minitans* spores was slightly lower than that of *T. harzianum*, but still high, being more than 90%. The percentage of germination was not statistically different between the control and the treatment with the ink-jet spray system (Figure 4). Conidia production of *C. minitans* was comparable between the two treatments after 21 days of incubation, with no statistically significant differences (Figure 4).

*G. mosseae* showed slower and overall lower germination than the two previously described fungi (Figure 5). However, the percentage of germination was not statistically different between the control and spores treated with the ink-jet system on all eight study dates.

The growth of *B. subtilis*, quantified by spectrophotometric measurement, was not statistically different between the control and the ink-jet system treatment on any of the examination dates (Figure 6). Growth of *P. fluorescens*, quantified by spectrophotometric measurement, was not statistically different between the control and treatment with the ink-jet system on all examination dates (Figure 7). Direct seeding of *P. fluorescens* with the ink-jet system onto a Petri dish containing LB substrate with the addition of agar demonstrated the viability of the cells after treatment and their ability to form new colonies (Figure 7).

The data from three different fungi from a taxonomic and ecophysiological point of view and from two bacteria belonging to two different phylums led to a similar conclusion: the spray application of organisms useful for biological control is compatible with the ink-jet system. All tests performed have shown that the fungal spores or bacterial cells remain active after treatment with the ink-jet system, with no statistically significant differences compared to the control. Of course, conclusions must be limited to the group of microorganisms used in the tests, and further experiments are needed to expand the list of microorganisms compatible with the ink-jet system.

### 3.2. Compatibility Test of TIJ with Herbicides

Both herbicides showed a strong effect depending on the application concentration (ANOVA, *p* < 0.01), whereas the effect of the type of application system (ANOVA, *p* = 0.85) and the interaction between concentration and application system were insignificant.

Glyphosate at the highest concentration caused plant death with more than 98% yellowing in *L. temulentum* in all three application systems (ST20, ST50, spray) (Figure 8). At a concentration of 10 μL mL^−1^, the yellowing percentage varied between 43.2% and 45.9% and then collapsed at lower concentrations. In the untreated control, the percentage of yellowing was very low (0.8%). At concentrations of 30 μL mL^−1^ and 10 μL mL^−1^, the percentage of leaf yellowing was slightly higher when applied with the ST20 system compared with ST50 and spray, although these differences were not statistically significant. For *A. fatua*, glyphosate at the highest concentration caused plant death in all three application systems (ST20, ST50, spray). At a concentration of 10 μL mL^−1^, the yellowing percentage varied from 29.1% to 34.2%. Glyphosate efficacy was also drastically reduced in *A. fatua* at lower concentrations, and the levels were only slightly higher than those of the control (3.4%). Additionally, in *A. fatua*, at the two highest concentrations of 30 μL ml^−1^ and 10 μL mL^−1^, the percentage of leaf yellowing was slightly higher when applied with the ST20 system compared to ST50 and spray. Again, as reported for *L. temulentum*, the differences were not statistically significant.

Cycloxidim had a slightly slower effect than glyphosate but was still highly concentration dependent. At the highest concentration, cycloxidim caused yellowing that ranged from 78.3% to 82.3% for *L. temulentum* and from 76.2% to 81.7% for *A. fatua* (Figure 9). The percentage of leaf yellowing was considerably reduced at a concentration of 3 μL mL^−1^ and was not different from the control at applications of 1 μL mL^−1^, 0.3 μL mL^−1^, and 0.3 μL mL^−1^. Similarly, for cycloxidim, the slight differences in leaf yellowing observed in *L. temulentum* and *A. fatua* by the different application systems were not statistically significant.

### 3.3. Test for Reducing Application Rate of Insecticides

Both insecticides showed a strong effect dependent on the application concentration (ANOVA, *p* < 0.01), while the effect of the type of application system (ANOVA, *p* = 0.89) and the interaction between concentration and application system was insignificant.

Abamectin at the highest concentration caused complete death of both larval species in all three application systems (ST20, ST50, spray) (Table 1). At a concentration of 0.3 μL mL^−1^, the percentage of dead larvae is very high in both species, and in the case of *G. melnella*, mortality was slightly higher with the application of ST20 and ST50 than with the spray. In contrast, mortality of *S. carnaria* at a concentration of 0.3 μL mL^−1^ was different among the three application systems. At a concentration of 0.1 μL mL^−1^, mortality was very low for both larval species, with no statistically significant differences between spray, ST20, and ST50. Finally, at concentrations of 0.03 μL mL^−1^ and 0.01 μL mL^−1^, no mortality was observed in either *G. melnella* or *S. carnaria* with all three application systems (Table 1).

Deltamethrin showed a concentration-dependent effect, as observed with abamectin, although efficacy was slightly higher. At the highest concentration, deltamethrin caused complete death of both larval species in all three application systems (ST20, ST50, spray) (Table 2). At a concentration of 0.1 μL mL^−1^, the percentage of dead larvae for both larval species is very high, exceeding 80%, but without any difference in results at the statistical level between the spray, ST20, and ST50 systems. At a concentration of 0.06 μL mL^−1^, mortality varied between 45% and 51% for *G. melnella* and between 35% and 43% for *S. carnaria*. Mortality was slightly higher in the case of application with the ST20 system, although these differences were not statistically significant. Finally, at the lowest concentrations (0.01 μL mL^−1^ and 0.006 μL mL^−1^), no mortality was observed for both *G. melnella* and *S. carnaria* with all three application systems.

### 3.4. Test for Reducing Application Rate of Beneficial Microbes by Modulating Droplet Size

Both beneficial microorganisms showed an effect that was strongly dependent on the application concentration (ANOVA, *p* < 0.01) and depended on the type of application system (ANOVA, *p* < 0.01).

In the case of *T. harzianum*, applications of the fungal spores at the lowest concentrations (1 × 10^3^ and 1 × 10^4^ mL^−1^) did not cause statistically significant differences in *L. perenne* growth compared with the untreated control (Figure 10). In contrast, at medium concentrations (1 × 10^5^ and 1 × 10^6^ mL^−1^), the application of *T. harzianum* caused a 20–25% increase in growth compared to untreated plants. In this case, no significant differences were observed between the spray, ST20, and ST50 systems (Figure 10). Finally, at the highest concentration (1 × 10^7^ mL^−1^), the application of fungal spores stimulated growth only in the spray system and in the ST50 application, while in the ST20 system, the growth of *L. perenne* was comparable to that of the control. This result is probably due to problems that occurred with the suspensions of *T. harzianum* spores at a concentration of 1 × 10^7^ mL^−1^ in ST20. In this configuration, the ST20 system operated for a few seconds and then stopped. The reverse ‘priming’ procedure, i.e., pressurizing the system with water, restored functionality but took several seconds when the system was used again. In contrast, no problems were observed with the ST50 configuration, indicating that the ST20 system was probably clogged at the highest concentration due to the high spore density. The same problem was observed at the 1 × 10^7^ mL^−1^ concentration with *B. subtilis*. It is important to note that no similar problems were observed with herbicides, insecticides, and fungicides even at the highest concentrations. In the case of *E. vesicaria*, the response to *T. harianum* application was very similar to *L. perenne*: no effect at the lowest concentrations, stimulation at medium and high concentrations. It is important to note that also for *E. vesicaria*, at a spore concentration of 1 × 10^7^ mL^−1^, the biostimulatory effect was detected only for the spray system and with the ST50 configuration, but not with ST20, probably due to the problems described above.

For *B. subtilis* applied to *L.perennial* grass, no growth-promoting or -inhibiting effects were detected at the lowest concentration, with values statistically similar to those of the control (1 × 10^3^ mL^−1^). Significant stimulation, up to +35% compared to the control, was observed at concentrations 1 × 10^4^, 1 × 10^5^, and 1 × 10^6^ mL^−1^ (Figure 11). At the highest concentration, the growth increase was less pronounced for the spray and ST50 system than for the control, while growth for the ST20 system was comparable to that of the control. This result can also be interpreted as an effect of the application problems observed with ST20 at the highest concentration for both fungi and bacteria. For *E. vesicaria*, the bacterium generally showed a less pronounced stimulatory effect on this species. In particular, no effect was detected at a concentration of 1 × 10^3^ mL^−1^, and the values were comparable to those of the control. At the concentrations of 1 × 10^4^, 1 × 10^5^, and 1 × 10^6^ mL^−1^, stimulation of up to 15% of the control concentration was observed (Figure 11). In all these cases, no statistically significant differences were observed between the spray, ST20, and ST50 applications. Finally, at the highest concentration (1 × 10^7^), an opposite result was observed: no effect in the ST20 configuration, but significant inhibition in the ST50 and spray configurations. This result can be interpreted with an antagonistic effect of the bacterium when it is present in very high density, an effect that has been previously reported in the literature [30]. This effect is less pronounced in the ST20 system, probably due to the clogging problem already described.

The data obtained show beyond doubt that the ink-jet system is capable of effectively applying various pesticides, including herbicides, insecticides, and biological control agents. In most trials, the ink-jet system in the ST20 and ST50 configurations showed results comparable to those of a pressure spray system. The only exception was the low efficacy of the ST20 system when applying *T. harzianum* spores and *B. subtilis* cells at the highest concentration (1 × 10^7^ mL^−1^). In this case, the ST20 system showed obvious clogging problems, probably caused by the high concentration of the liquid suspension. However, from an application perspective, this should not be a major problem as these products are typically used at lower concentrations of 1 × 10^3^ to 1 × 10^6^ spores per mL. Overall, the tests conducted have shown that the ST20 and ST50 spray and ink-jet systems are similarly effective. Therefore, at this stage of the study, it cannot be said that the ST20 or ST50 ink-jet system can significantly reduce pesticide dosage compared to a standard pressure system. Clearly, our conclusions must be limited to the group of active substances and microorganisms used in the trials, and further testing is needed. In particular, additional series of experiments could be carried out in which the spray volumes rather than the concentrations of the chemicals are changed.

### 3.5. Application of TIJ System to Microgreens

As for the test of droplet distribution with hydrosensitive papers, application with ST50 resulted in complete coverage of the surface, while with ST20, the surface affected by the color change was only 38%. With the spray system, the coverage of the color change was between ST50 and ST20 (Figure 12). Image analysis also showed that with ST20, the blue colored surface followed the lines of the path of the ink-jet system, but without covering the entire surface of interest. Observations made during the tests indicate that the ST20 system suffered from a strong drift effect even under laboratory conditions and thus at high air stability due to the complete absence of wind and air currents. For the spray system, coverage was lower than for the ST50 at the same height and application rate. This result could be due to the larger size of the droplets, which did not drift but could not completely cover the treated area (Figure 12). In the second experiment, the distance from the support surface had a statistically significant effect on the percentage of blue colored surface, which decreased with increasing distance (Figure 13). For the ST50 system, coverage was complete at 5 and 10 cm, and slightly less at 15 and 20 cm, but values were always above 80%. For the ST20 system, the area affected by the color change was 55% at 5 cm, with a drastic reduction as the distance increased and values as low as 7% at 20 cm. This result is not surprising given the first experiment and the considerations of the drift susceptibility of very fine droplets even when there was no wind. At a distance of 20 cm, the area in contact with water was actually very small for the same spray volume. For the standard spray, the reduction in coverage as a function of height was statistically significant, but less pronounced than for the ST20. Since drift is not relevant in this case, the reduction could be due to thinning of the droplets with distance, which appears larger when analyzed visually at a distance of 20 cm. In any case, the spray system determined lower coverage values than the ST50 at all heights evaluated (Figure 13).

Regarding the application of beneficial microbes to microgreens, *T. harzianum* had an effect depending on the application height (ANOVA, *p* < 0.01) and by the type of application system (ANOVA, *p* < 0. 01). Application of the spores with the spray and ST50 system resulted in statistically significant differences compared with the control with stimulation ranging from +14% to +21% (Figure 14). However, no differences were observed between the spray treatment and the ST50 system in terms of application level. Conversely, application of *T. harzianum* stimulated lettuce growth by 13% when treated with the ST20 system at 10 cm height, but no effect was observed compared to the control when spores were applied at 15 and 20 cm height. This result is probably due to the limited ability of the ST20 system to reach the target (the leaf surface) when the distance is more than 10 cm, as shown by the tests with the water-sensitive paper sheet. As far as rocket is concerned, the results are consistent with those described for lettuce. In general, the application of *T. harzianum* resulted in stimulation compared to the control, reaching +29% for spray, +27% for ST50, and +8% for ST20 (Figure 14). As noted in lettuce, biostimulation in rocket was comparable with the ST50 and spray systems and was independent of application height. Conversely, biostimulation with the ST20 system was lower at an application height of 10 cm and absent at application heights of 15 and 20 cm. The lower biostimulation in rocket compared to lettuce at an application height of 10 cm could be due to the smaller stature of the first species.

## 4. Conclusions

The data obtained during the tests with water-sensitive papers showed that the ST20 system, which produces droplets with a diameter of ~20 μm, is subject to strong drift even at distances of 10 cm or more from the firing surface. It should also be noted that the experiments were conducted in the laboratory under complete windless conditions. It is therefore easy to assume that even under sheltered growing conditions, i.e., when the air masses are relatively stable, as may occur in greenhouse conditions, the problem of drift is still considerable. The problem of very fine droplets is well known in the literature [31], and although they may offer advantages, such as better coverage, reduced drip losses, and possibly stomata penetration, these are undoubtedly negated by the effects of drift. In this context, the use of ST20 with varying potential did not provide any benefits. The ST50 system showed high coverage efficiency in the 5–20 cm range tested. Consequently, subsequent tests with both herbicides and beneficial microorganisms have shown that this system can be used for growing microgreens. In addition, from an application standpoint, the integration of the ink-jet system into DEMO was successful and could be adapted to a microgreens growing systems with small plants and heights less than 15–20 cm and limited leaf area. However, extending it to other cropping systems is a major challenge. First of all, in its current configuration, the system can only apply very small amounts of liquid. Currently, a gradual reduction in the amounts used for pesticide treatment is observed in agricultural production. Specifically, the amounts used range from 200–500 L/ha to over 1500 L/ha. The ST50 system is currently not capable of even approaching the lower end of this spectrum. With this in mind, developing a system that uses ST50 in parallel could be an opportunity. Another option to be explored could be the development of a delivery system capable of producing larger droplets in the spectrum between 100 and 200 μm. This system could be capable of delivering larger volumes and therefore could be compared to systems currently used for pesticide distribution (e.g., mechanical and pneumatic sprayers) in other cropping systems (protected and field crops). In conclusion, this study, for the first time, demonstrated a potential application of the ink-jet technology for the distribution of agrochemicals. Future research will be needed to evaluate the real applicability of this technology to different cultivation systems.

## Figures and Tables

**Figure 1 sensors-23-03053-f001:**
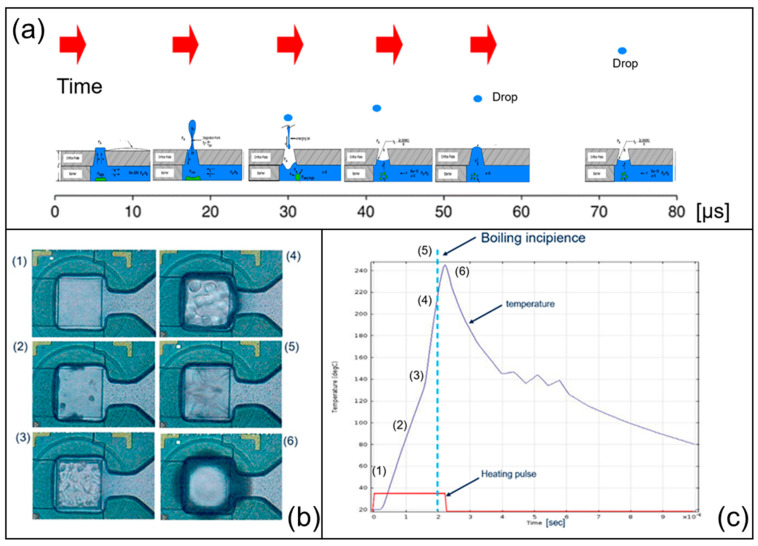
(**b**) thin-film heater with dimensions of 50 × 50 μm made of TaSin. Pulse power Q = 15 W, pulse width *τ* = 2.5 μs, turn on energy TOE = 0.0000375 J. The temporal evolution with bubble growth is presented (**a**). Thermal ink-jet jetting mechanism in relation with pulse heating (**b**), temperature, pressure, and bubble volume (**c**).

**Figure 2 sensors-23-03053-f002:**
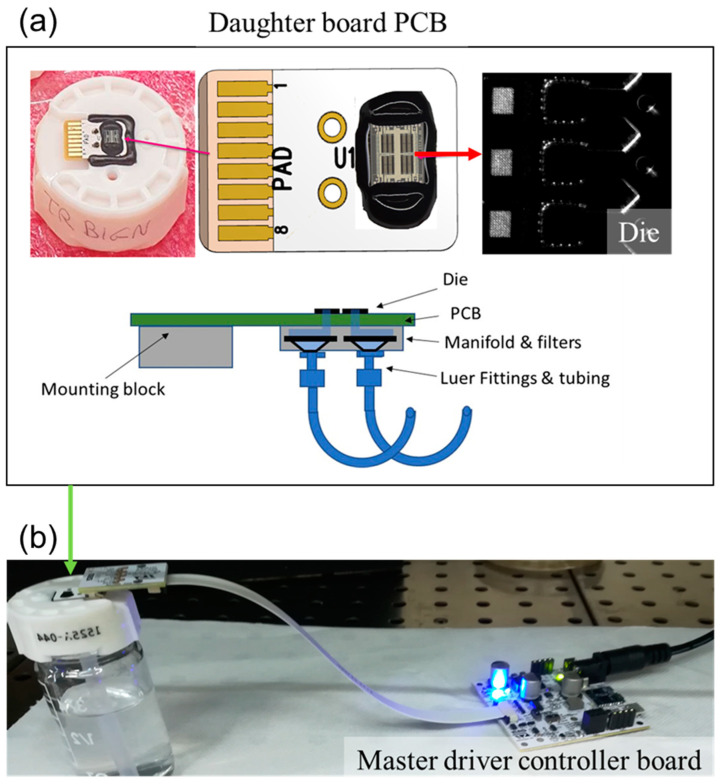
Structure of the Printed Circuit Board (PCB) with microscopic magnification of the die (red arrow) attached on one side of the PCB (**a**); the system set-up assembled and composed by the daughter board PCB and the master driver controller board with power cable (**b**).

**Figure 3 sensors-23-03053-f003:**
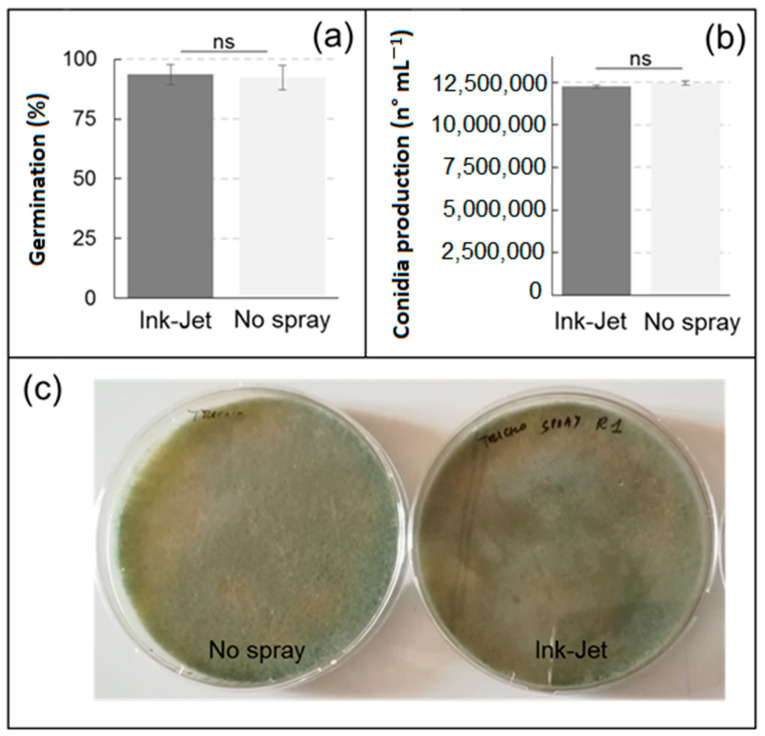
Spore germination (**a**) and conidia production (**b**) of *Trichoderma harzianum* after treatment with the ink-jet spray system. Data are means ± standard deviation, ns = statistically insignificant difference (Duncan’s test). (**c**) images of *Trichoderma harzianum* cultures on PDA obtained after seven days of incubation with control spores (**left**) or applied following treatment with ink-jet system (**right**).

**Figure 4 sensors-23-03053-f004:**
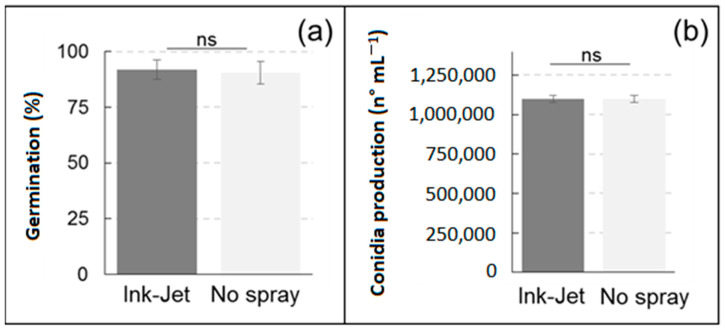
Spore germination (**a**) and conidia production (**b**) of *Coniothyrium minitans* in relation to treatment with the ink-jet spray system. Data are means ± standard deviation, ns = statistically insignificant difference (Duncan’s test).

**Figure 5 sensors-23-03053-f005:**
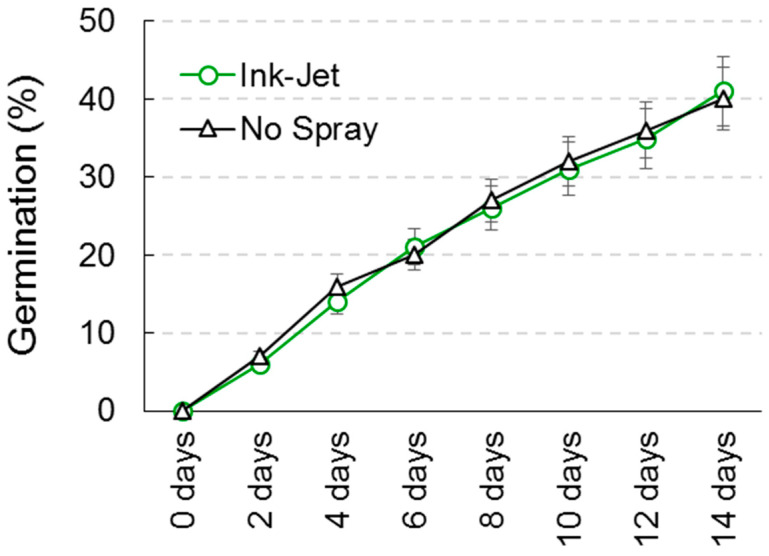
Germination of *Glomus mosseae* spores in relation to ink-jet spray system treatment. Data are means ± standard deviation.

**Figure 6 sensors-23-03053-f006:**
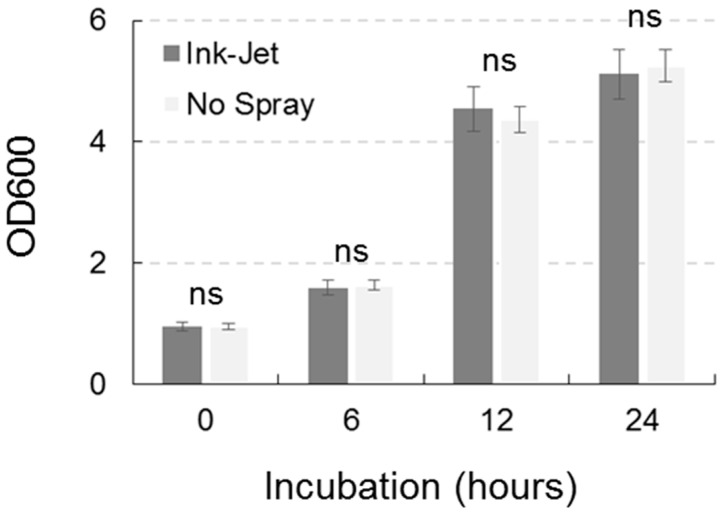
Growth of *Bacillus subtilis* in relation to treatment with ink-jet spray system. Data are means ± standard deviation, ns = statistically insignificant difference (Duncan’s test).

**Figure 7 sensors-23-03053-f007:**
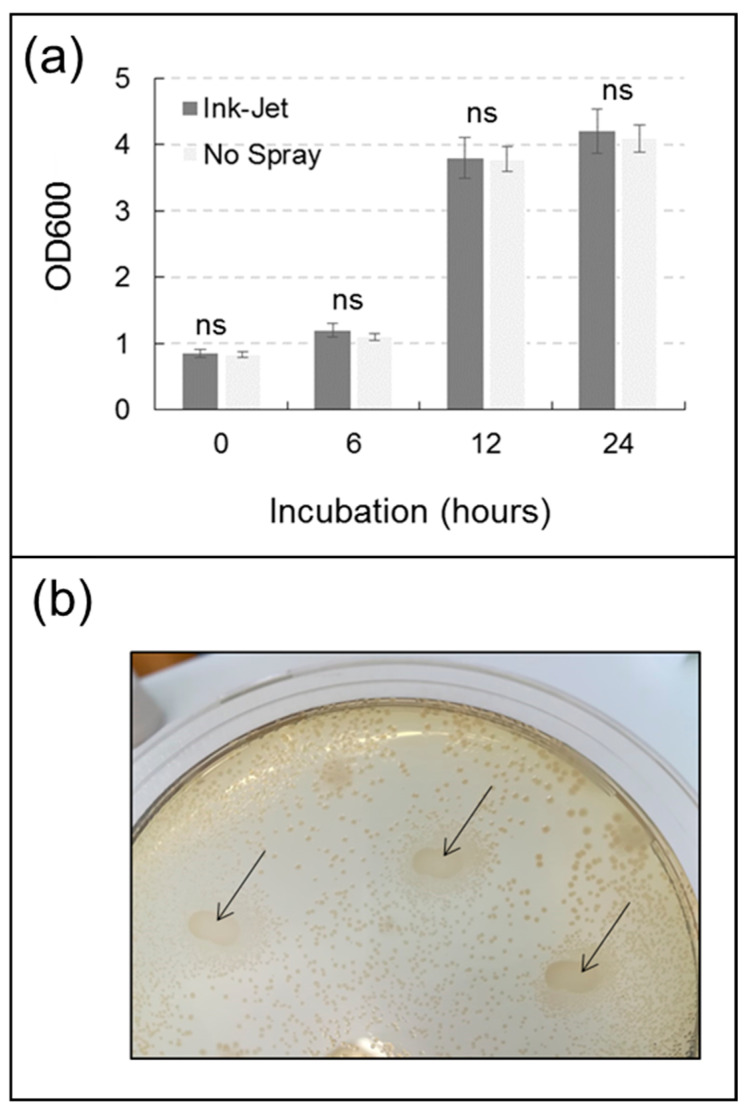
Growth of *Pseudomonas fluorescens* in relation to treatment with ink-jet spray system (**a**). Data are means ± standard deviation, ns = statistically insignificant difference (Duncan’s test). Image of a detail of a Petri dish containing LB substrate with the addition of agar and on which a suspension of *Pseudomonas fluorescens* cells has been sprayed with the ink-jet system at three points (black arrows) (**b**).

**Figure 8 sensors-23-03053-f008:**
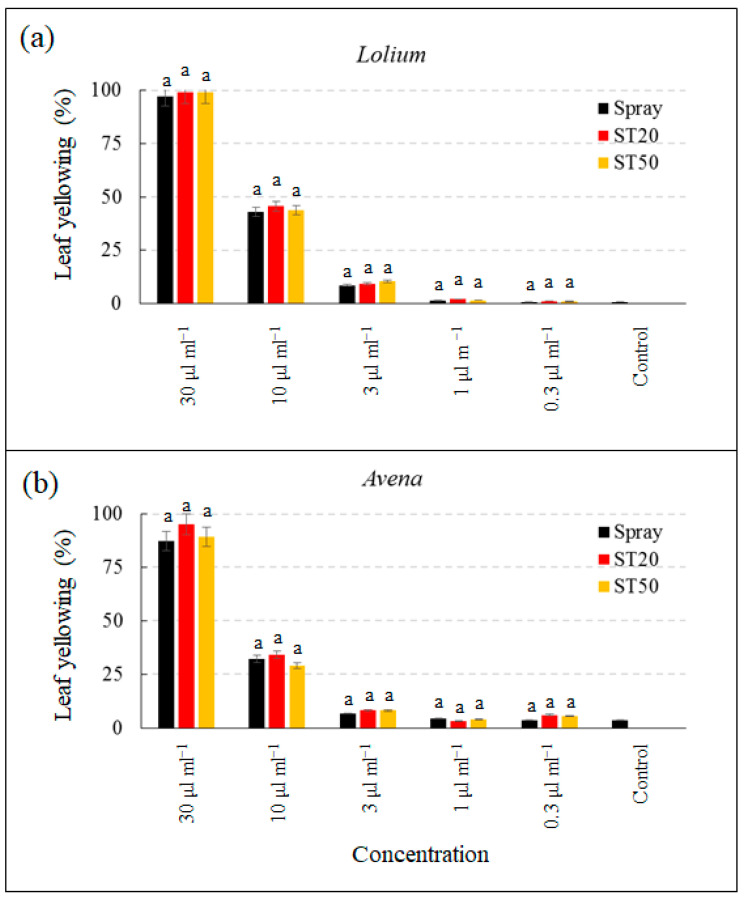
Percentage of yellowed leaf area of *Lolium temulentum* (**a**) and *Avena fatua* (**b**) treated with glyphosate at five concentrations with spray system, ST20, and ST50. Data are means ± standard deviation; different letters indicate statistically significant differences within concentration (Duncan’s test).

**Figure 9 sensors-23-03053-f009:**
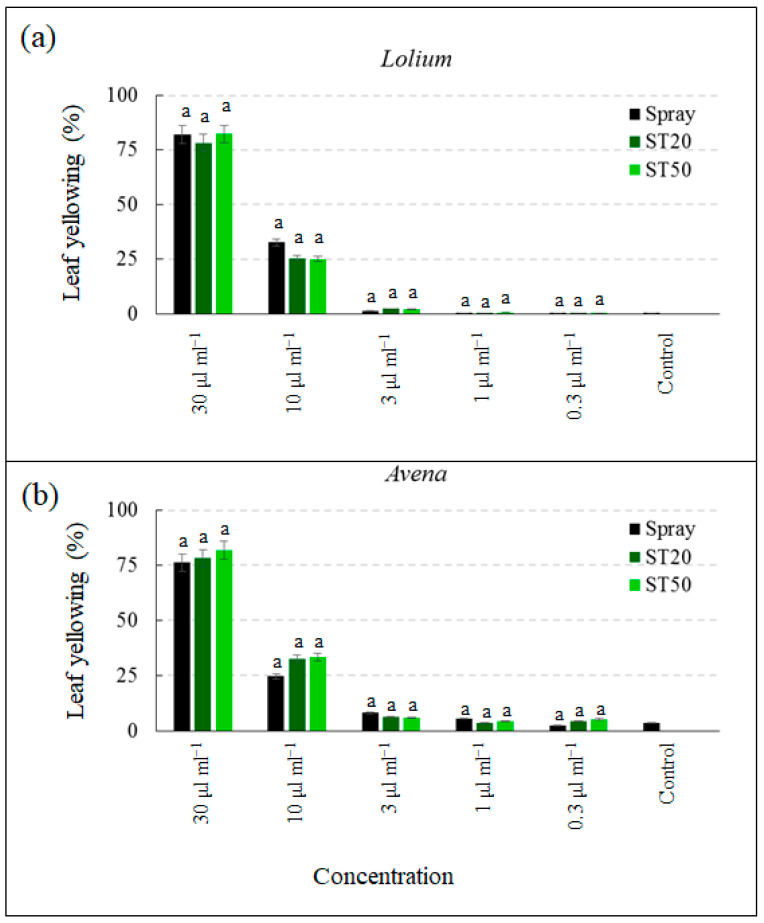
Percentage of yellowed leaf area of *Lolium temulentum* (**a**) and *Avena fatua* (**b**) treated with cicloxidim at five concentrations with spray system, ST20, and ST50. Data are means ± standard deviation; different letters indicate statistically significant differences within concentration (Duncan’s test).

**Figure 10 sensors-23-03053-f010:**
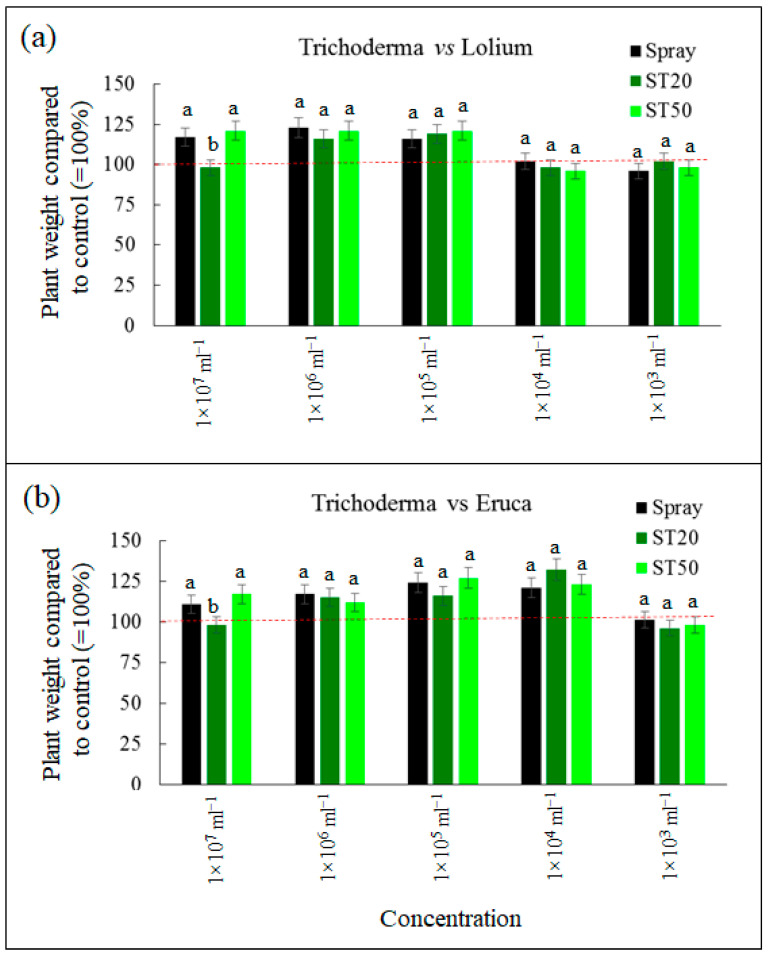
Growth of *Lolium perenne* (**a**) and *Eruca vesicaria* (**b**) treated with *Trichoderma harzianum* at five concentrations with spray system, ST20, and ST50. Data are means ± standard deviation; different letters indicate statistically significant differences within concentration (Duncan’s test). The red line indicates the growth of the untreated control.

**Figure 11 sensors-23-03053-f011:**
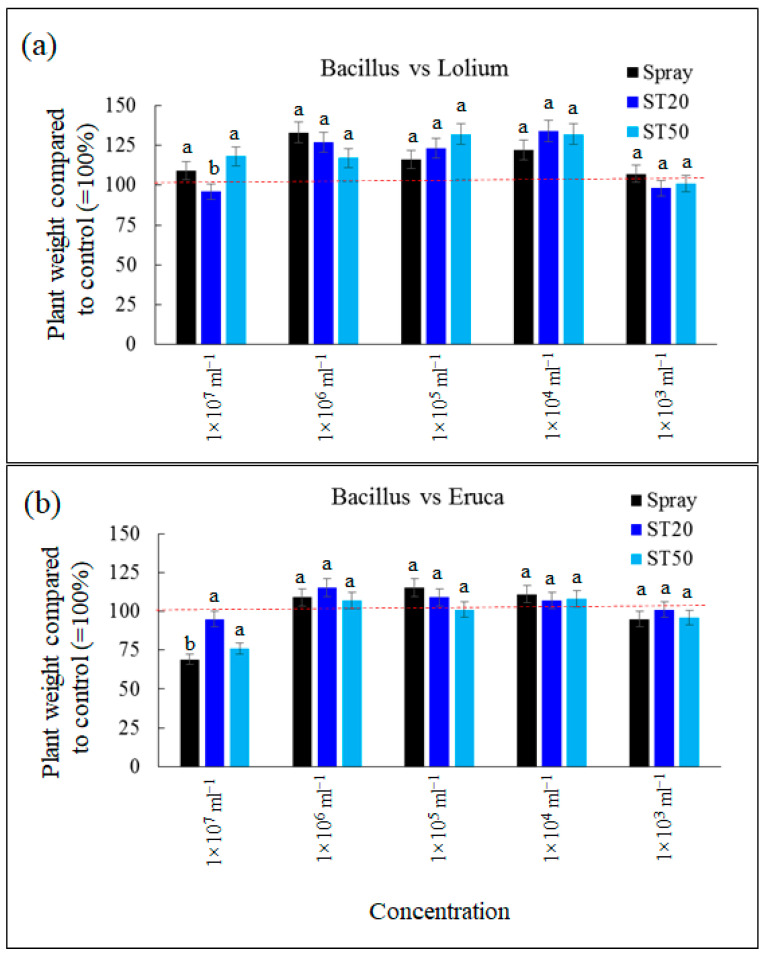
Growth of *Lolium perenne* (**a**) and *Eruca vesicaria* (**b**) treated with *Bacillus subtilis* at five concentrations with spray system, ST20, and ST50. Data are means ± standard deviation; different letters indicate statistically significant differences within concentration (Duncan’s test). The red line indicates the growth of the untreated control.

**Figure 12 sensors-23-03053-f012:**
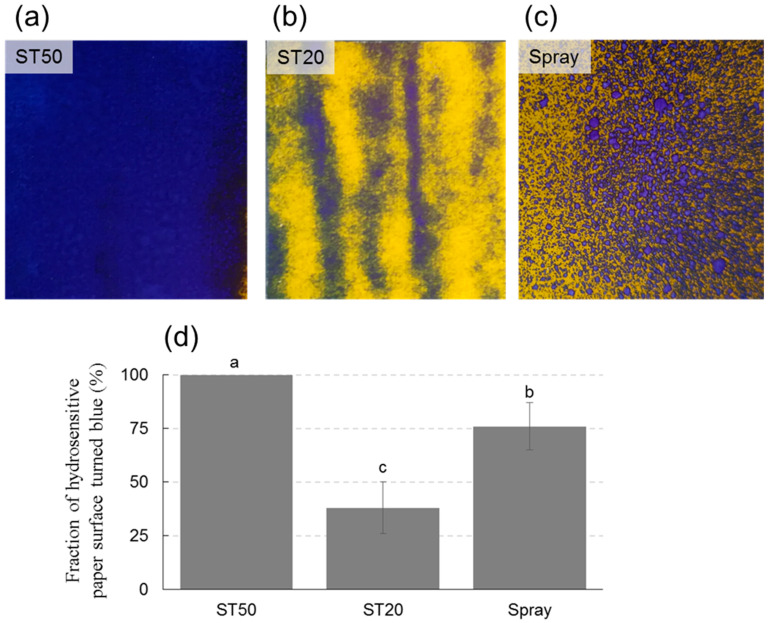
Percentage of hydrosensitive paper surface turned blue, and therefore reached by water, following treatment with the ST50 (**a**), ST20 (**b**) and Spray systems (**c**) applied at 10 cm from the supporting surface. Data are means ± standard deviation; different letters indicate statistically significant differences (Duncan’s test) (**d**). The photos show representative examples of the water sensitive papers 5 min after treatment. Note the large droplets in the spray treatment (**c**).

**Figure 13 sensors-23-03053-f013:**
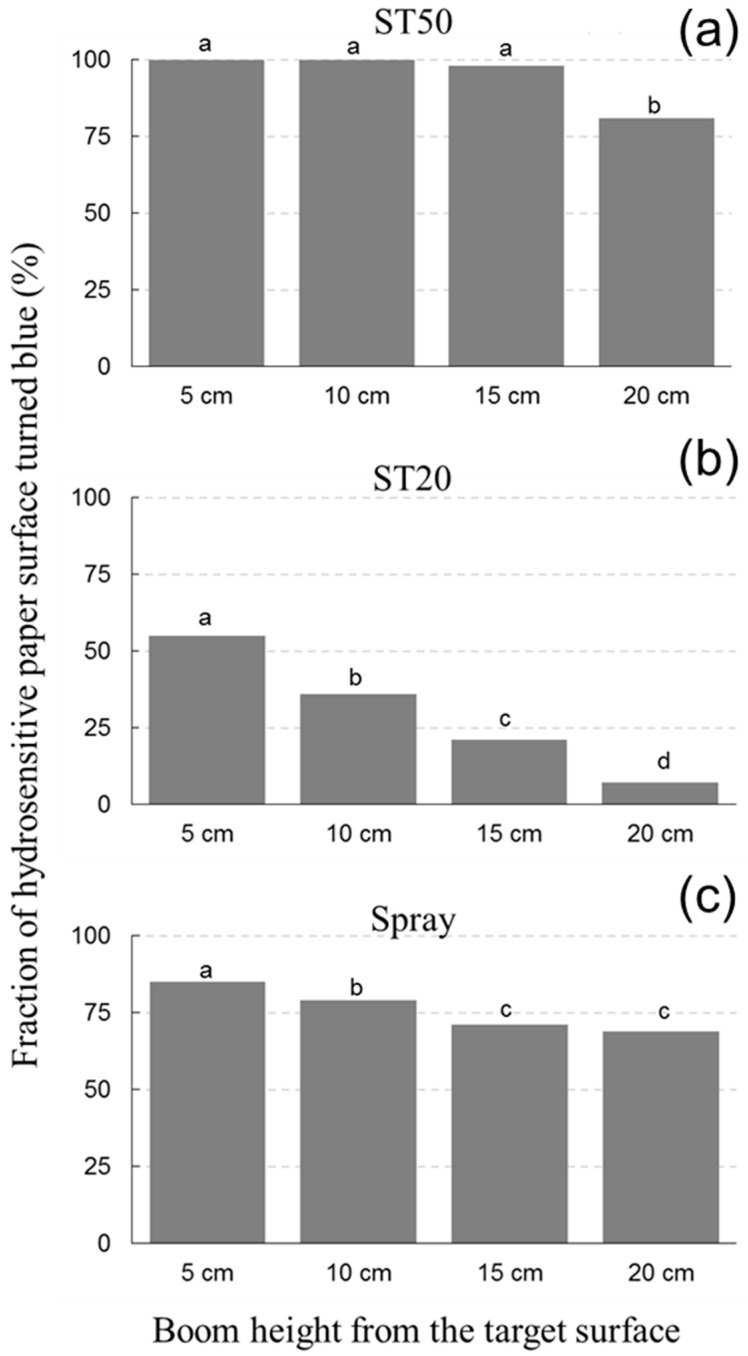
Percentage of hydrosensitive paper surface turned blue, and therefore reached by water, following treatment with the ST50 (**a**), ST20 (**b**), and Spray (**c**) systems at 5, 10, 15, and 20 cm from the supporting surface. Data are means of three replicates; different letters indicate statistically significant differences (Duncan’s test).

**Figure 14 sensors-23-03053-f014:**
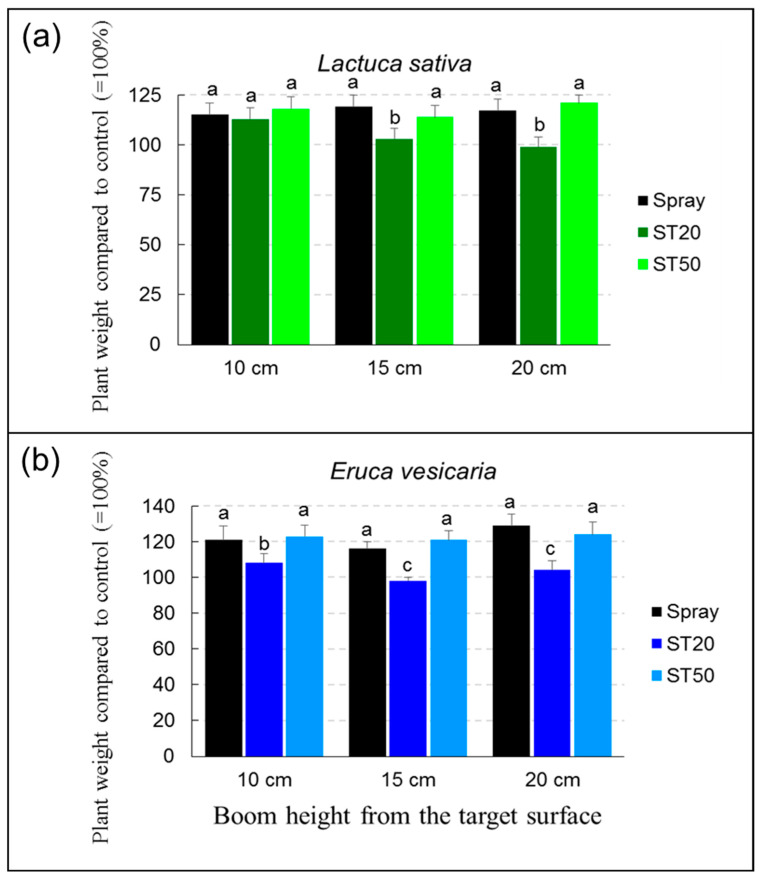
Growth of *Lactuca sativa* (**a**) and *Eruca vesicaria* (**b**) treated with *Trichoderma harzianum* at three application heights (10, 15, and 20 cm) with spray system, ST20, and ST50 integrated in the DEMO. Data are means ± standard deviation; different letters indicate statistically significant differences (Duncan’s test).

**Table 1 sensors-23-03053-t001:** Percentage of *G. mellonella* and *S. carnacia* larvae that died following treatment with five concentrations of abamectin by spray system, ST20, and ST50. Data are means ± standard deviation; different letters indicate statistically significant differences within concentration (Duncan’s test).

Active Compound and Commercial Name	Species	Concentration	Spray	ST20	ST50
Abamectin (VERTIMEC)	*G. mellonella*	1 μL mL^−1^	100 ± 0 a	100 ± 0 a	100 ± 0 a
		0.3 μL mL^−1^	83 ± 6 b	91 ± 6 a	90 ± 9 a
		0.1 μL mL^−1^	5 ± 3 a	9 ± 5 a	7 ± 5 a
		0.03 μL mL^−1^	0 ± 0 a	0 ± 0 a	0 ± 0 a
		0.01 μL mL^−1^	0 ± 0 a	0 ± 0 a	0 ± 0 a
	*S. carnaria*	1 μL mL^−1^	100 ± 0 a	100 ± 0 a	100 ± 0 a
		0.3 μL mL^−1^	78 ± 8 a	81 ± 9 a	79 ± 11 a
		0.1 μL mL^−1^	0 ± 0 a	0 ± 0 a	0 ± 0 a
		0.03 μL mL^−1^	0 ± 0 a	0 ± 0 a	0 ± 0 a
		0.01 μL mL^−1^	0 ± 0 a	0 ± 0 a	0 ± 0 a

**Table 2 sensors-23-03053-t002:** Percentage of *G. mellonella* and *S. carnacia* larvae that died following treatment with five concentrations of deltametrin by spray system, ST20, and ST50. Data are means ± standard deviation; different letters indicate statistically significant differences within concentration (Duncan’s test).

Active Compound and Commercial Name	Species	Concentration	Spray	ST20	ST50
Deltametrine (DECIS)	*G. mellonella*	0.6 μL mL^−1^	100 ± 0 a	100 ± 0 a	100 ± 0 a
		0.1 μL mL^−1^	95 ± 3 a	91 ± 9 a	87 ± 8 a
		0.06 μL mL^−1^	45 ± 3 a	48 ± 8 a	51 ± 11 a
		0.01 μL mL^−1^	0 ± 0 a	0 ± 0 a	0 ± 0 a
		0.006 μL mL^−1^	0 ± 0 a	0 ± 0 a	0 ± 0 a
	*S. carnaria*	0.6 μL mL^−1^	100 ± 0 a	100 ± 0 a	100 ± 0 a
		0.1 μL mL^−1^	85 ± 2 a	81 ± 9 a	86 ± 8 a
		0.06 μL mL^−1^	35 ± 12 a	43 ± 13 a	38 ± 9 a
		0.01 μL mL^−1^	0 ± 0 a	0 ± 0 a	0 ± 0 a
		0.006 μL mL^−1^	0 ± 0 a	0 ± 0 a	0 ± 0 a

## Data Availability

Data are available upon request.

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
