# Peer review of "Structure, Functionality, Compatibility with Pesticides and Beneficial Microbes, and Potential Applications of a New Delivery System Based on Ink-Jet Technology"

_sensors, 2023, doi:10.3390/s23063053_

Round 1
Reviewer 1 Report
The article is presented in an articulate manner
it is recommended :
- Revise the format of the article according to the publisher's recommendations for author name sections, placement of graphs, table styles, nomenclature, and letter styles for chemical formulas, units, among others.
- Provide the e-mail addresses of the rest of the authors.
- Use capital letters at the beginning of sentences.
- Units of measurement are in italics, check throughout the document.
- The distribution of the figures should be organised according to your best judgement by means of panels which will be named a), b), c) ..... and described in the corresponding caption and in the text of the article.
- Keep the style first the full name and then the acronym in brackets, then only use the acronym.
- Chemical formulae should be written in italics and, if necessary, numbered as an equation.
- Keep the descriptions of the figures both in the caption and in the text of the article.
- A concluding section is needed where the general conclusions of the article are addressed.

Author Response
Reviewer #1
The article is presented in an articulate manner it is recommended:
- Revise the format of the article according to the publisher's recommendations for author name sections, placement of graphs, table styles, nomenclature, and letter styles for chemical formulas, units, among others.
DONE. We thank the referee for the positive and constructive comments. We performed a detailed check on the format and, following the attached pdf file of the referee, we modified the tables and figures as requested.
- Provide the e-mail addresses of the rest of the authors.
DONE. We added the email of all authors.
- Use capital letters at the beginning of sentences.
DONE.
- Units of measurement are in italics, check throughout the document.
DONE.
- The distribution of the figures should be organised according to your best judgement by means of panels which will be named a), b), c) ..... and described in the corresponding caption and in the text of the article.
DONE. Thank you for the suggestion, we amended all figures and the associated captions with description of the different panels.
- Keep the style first the full name and then the acronym in brackets, then only use the acronym.
DONE.
- Chemical formulae should be written in italics and, if necessary, numbered as an equation.
DONE.
- A concluding section is needed where the general conclusions of the article are addressed
DONE. Following the referee section, we add a Conclusion section at the end of the manuscript as follows: ““4. Conclusions.
The data obtained during the tests with water-sensitive papers showed that the ST20 system, which produces droplets with a diameter of ~20 μm, is subject to strong drift even at distances of 10 cm or more from the firing surface. It should also be noted that the experiments were conducted in the laboratory under complete windless conditions. It is therefore easy to assume that even under sheltered growing conditions, i.e., when the air masses are relatively stable, as may occur in greenhouse conditions the problem of drift is still considerable. The problem of very fine droplets is well known in the literature [31], and although they may offer advantages, such as better coverage, reduced drip losses, and possibly stomata penetration, these are undoubtedly negated by the effects of drift. In this context, the use of ST20 with varying potential did not provide any benefits. The ST50 system showed high coverage efficiency in the 5-20 cm range tested. Consequently, subsequent tests with both herbicides and beneficial microorganisms have shown that this system can be used for growing microgreens. In addition, from an application standpoint, the integration of the Ink-Jet system into DEMO was successful and could be adapted to microgreens growing systems with small plants and heights less than 15-20 cm and limited leaf area. However, extending it to other cropping systems is a major challenge. First of all, in its current configuration, the system can only apply very small amounts of liquid. Currently, a gradual reduction in the amounts used for pesticide treatment is observed in agricultural production. Specifically, the amounts used range from 200-500 l/ha to over 1500 l/ha. The ST50 system is currently not capable of even approaching the lower end of this spectrum. With this in mind, developing a system that uses ST50 in parallel could be an opportunity. Another option to be explored could be the development of a delivery system capable of producing larger droplets in the spectrum between 100 and 200 μm. This system could be capable of delivering larger volumes and therefore could be compared to systems currently used for pesticide distribution (e.g., mechanical and pneumatic sprayers) in other cropping systems (protected and field crops). In conclusion, this study, for the first time, demonstrated a potential application of the ink-jet technology for the distribution of agrochemical. Future research will be needed to evaluate the real applicability of this technology to different cultivation systems”.
Reviewer 2 Report
Introduction: the reader should be contextualized in relation to the logical order of the article.
Review of Literature: The article shows the entire experimentation process and the conclusions, however, a clear comparison between the proposed solution with others existing in the literature cannot be evidenced, a section focused on the systematic review of the literature is necessary, I recommend the inclusion of these articles that enrich the review:
- Ukhurebor, K.E., Aigbe, U.O., Onyancha, R.B., & Adetunji, C.O. (2021). Climate change and pesticides: their consequence on microorganisms. Microbial Rejuvenation of Polluted Environment: Volume 3, 83-113.
- Take, M., Choudhary, R., & Patidar, R. (2023). Chapter-1 Nano Urea-A Bliss for Agriculture or Not. Recent Trends in Agriculture, 1.
- Carlos, A. D. J., Estrada, L. R., Augusto, C. R. C., Patricia, A. C. P., Alberto, P. M. M., Enrique, R. G. R., ... & Andrés, C. M. C. (2020). Monitoring system of environmental variables for a strawberry crop using IoT tools. Proceeds Computer Science, 170, 1083-1089.
- Vasilchenko, A. V., Poshvina, D. V., Semenov, M. V., Timofeev, V. N., Iashnikov, A. V., Stepanov, A. A., ... & Vasilchenko, A. S. (2023). Triazoles and Strobilurin Mixture Affects Soil Microbial Community and Incidences of Wheat Diseases. Plants, 12(3), 660.
- Fan, X., Zhao, M., Wen, H., Zhang, Y., Zhang, Y., Zhang, J., & Liu, X. (2023). Enhancement degradation efficiency of pyrethroid-degrading esterase (Est816) through rational design and its application in bioremediation. Chemosphere, 138021.
- Barrios-Ulloa, A., Ariza-Colpas, P. P., Sánchez-Moreno, H., Quintero-Linero, A. P., & De la Hoz-Franco, E. (2022). Modeling radio wave propagation for wireless sensor networks in vegetated environments: A systematic literature review. Sensors, 22(14), 5285.
Author Response
Reviewer #2
Introduction: the reader should be contextualized in relation to the logical order of the article. DONE. The Introduction has been reorganized following the suggestion. In detail, we added several references to compare better the new system with existing technologies including IoT system that was not mentioned in the early manuscript version. Overall, the text was modified and integrated as follows: “Smart and precision farming based on robotics, machine automation, location technology and advanced data analysis including Internet Of Things [2, 3], has been widely studied to increase the efficiency in the use of fertilizers and pesticides to maximize crop yields and reduce production losses due to pest and diseases [4, 5]”.
Moreover, a sentence about previous application of ink-jet system was add as follows: “Recently, inkjet printing emerged as forefront for biosensor manufacturing approach, including point-of-care diagnostic biosensors [13, 14]”.
Overall, we used three of the suggested references i.e., Carlos et al. (2020). Proceeds Computer Science, 170, 1083-1089. Vasilchenko et al. (2023). Plants, 12(3), 660; Barrios-Ulloa et al. (2022). Sensors, 22(14), 5285; Fan et al. 2023. Chemosphere, 138021.
Reviewer 3 Report
The manuscript describes an exciting application of inkjets: delivery of pesticides and microbes to plants. This idea is novel; however, authors could add some references to show similar applications of inkjets, e.g., in biosensors. A primary concern about the manuscript is that it includes long descriptions and texts. It should be more compressed as a research paper focusing on the most important backgrounds, methods and results. Especially the introduction part reads like a book chapter or a general description of the topic. The manuscript includes 24 references which authors could improve, e.g., with more references in the introduction and discussion parts. Further to that, the conclusions part needs to be included. Authors should follow the classical structure of introduction, materials and methods, results and discussions and conclusions. Some parts of the materials and methods would fit into the introduction, e.g. lines 276-272.
Overall, the manuscript is interesting, with well-described research and well-presented data underlined with appropriate statistical analysis. Authors should consider abbreviating some parts or adding specific results to supplementary.
Line 187: Is there any reference to the viability test method?
Author Response
Reviewer #3
The manuscript describes an exciting application of inkjets: delivery of pesticides and microbes to plants. This idea is novel; however, authors could add some references to show similar applications of inkjets, e.g., in biosensors.
DONE. Thank you for the suggestion, we integrated the Introduction with the following sentence: “Recently, inkjet printing emerged as forefront for biosensor manufacturing approach, including point-of-care diagnostic biosensors [13, 14]”.
A primary concern about the manuscript is that it includes long descriptions and texts. It should be more compressed as a research paper focusing on the most important backgrounds, methods and results. Especially the introduction part reads like a book chapter or a general description of the topic. The manuscript includes 24 references which authors could improve, e.g., with more references in the introduction and discussion parts.
DONE. The Introduction has been reorganized following the suggestion. In detail, we add several reference to better compared the new system with existing technologies including IoT system that was not mentioned in the early manuscript version. Overall, the text was modified and integrated as follows: “Smart and precision farming based on robotics, machine automation, location technology and advanced data analysis including Internet Of Things [2, 3], has been widely studies to increase the efficiency in the use of fertilizers and pesticides to maximize crop yields and reduce production losses due to pest and diseases [4, 5]”. Other minor correction was made, please check the track changes version for details.
Further to that, the conclusions part needs to be included. Authors should follow the classical structure of introduction, materials and methods, results and discussions and conclusions.
DONE. Following the referee suggestion, we separated including a new Conclusion section. Here the text of the new conclusion:
“4. Conclusions.
The data obtained during the tests with water-sensitive papers showed that the ST20 sys-tem, which produces droplets with a diameter of ~20 μm, is subject to strong drift even at distances of 10 cm or more from the firing surface. It should also be noted that the experiments were conducted in the laboratory under complete windless conditions. It is there-fore easy to assume that even under sheltered growing conditions, i.e., when the air mass-es are relatively stable, as may occur in greenhouse conditions the problem of drift is still considerable. The problem of very fine droplets is well known in the literature [31], and although they may offer advantages, such as better coverage, reduced drip losses, and possibly stomata penetration, these are undoubtedly negated by the effects of drift. In this context, the use of ST20 with varying potential did not provide any benefits. The ST50 sys-tem showed high coverage efficiency in the 5-20 cm range tested. Consequently, subsequent tests with both herbicides and beneficial microorganisms have shown that this sys-tem can be used for growing microgreens. In addition, from an application standpoint, the integration of the Ink-Jet system into DEMO was successful and could be adapted to microgreens growing systems with small plants and heights less than 15-20 cm and limited leaf area. However, extending it to other cropping systems is a major challenge. First of all, in its current configuration, the system can only apply very small amounts of liquid. Currently, a gradual reduction in the amounts used for pesticide treatment is observed in agricultural production. Specifically, the amounts used range from 200-500 l/ha to over 1500 l/ha. The ST50 system is currently not capable of even approaching the lower end of this spectrum. With this in mind, developing a system that uses ST50 in parallel could be an opportunity. Another option to be explored could be the development of a delivery system capable of producing larger droplets in the spectrum between 100 and 200 μm. This sys-tem could be capable of delivering larger volumes and therefore could be compared to systems currently used for pesticide distribution (e.g., mechanical and pneumatic spray-ers) in other cropping systems (protected and field crops). In conclusion, this study, for the first time, demonstrated a potential application of the ink-jet technology for the distribution of agrochemical. Future research will be needed to evaluate the real applicability of this technology to different cultivation systems”.
Some parts of the materials and methods would fit into the introduction, e.g. lines 276-272.
DONE. Ok, we move this sentence in the Introduction as follows: “In a second step, we evaluated the potential of ink-jet technology for agrochemical reduction by investigating droplet size optimization for herbicides and insecticides that, globally, are economically important crop protection products”.
Overall, the manuscript is interesting, with well-described research and well-presented data underlined with appropriate statistical analysis. Authors should consider abbreviating some parts or adding specific results to supplementary.
The reviewer is surely right when he says that the manuscript is long. In this case, however, the length is justified by the need to report a detailed description of the system used and by the amount of experiments and data needed to test and validate it. In fact, in the original version many details are reported as supplementary material (four figures and four tables). For these reasons, we think, even taking into account that Sensors Journal has no space limitations, that it is more useful for the reader to keep the selected figures in the main text.
Line 187: Is there any reference to the viability test method?
DONE. Ok, we add the reference “Barile et al. (2007). Saponins from Allium minutiflorum with antifungal activity. Phytochemistry, 68(5), 596-603” which reports the methodology used for tests with fungi.
Reviewer 4 Report
This article is an innovative application of TIJ to the delivery system. It assessed the compatibility of TIJ technology with a range of pesticides and beneficial microorganisms and investigated the feasibility of using TIJ technology in micro vegetable production system.
Table 2 has no headings.
The summary at the end of the article gives a good account of the shortcomings of the study and an outlook for the future, without a good and detailed summary of the previous experimental content alone.
The potential applications of the ink-jet system in agriculture are numerous. For example, the system can be used for targeted pest management, delivering pesticides directly to the pest-infested area while avoiding non-target organisms. The system can also be used for the delivery of beneficial microbes, such as mycorrhizal fungi, which can enhance plant growth and nutrient uptake. Additionally, the ink-jet system can be used for the delivery of plant hormones or other biostimulants, which can improve plant growth and resilience to stress. The author can make subsequent studies in future.
Author Response
Reviewer #4
This article is an innovative application of TIJ to the delivery system. It assessed the compatibility of TIJ technology with a range of pesticides and beneficial microorganisms and investigated the feasibility of using TIJ technology in micro vegetable production system.
We would like to thank the reviewer for his positive comment and appreciation of the effort done for this work.
Table 2 has no headings.
The table was adjusted in the new version of the manuscript.
The summary at the end of the article gives a good account of the shortcomings of the study and an outlook for the future, without a good and detailed summary of the previous experimental content alone.
The potential applications of the ink-jet system in agriculture are numerous. For example, the system can be used for targeted pest management, delivering pesticides directly to the pest-infested area while avoiding non-target organisms. The system can also be used for the delivery of beneficial microbes, such as mycorrhizal fungi, which can enhance plant growth and nutrient uptake. Additionally, the ink-jet system can be used for the delivery of plant hormones or other biostimulants, which can improve plant growth and resilience to stress. The author can make subsequent studies in future.
The conclusions part was adjusted as follows: "
The data obtained during the tests with water-sensitive papers showed that the ST20 system, which produces droplets with a diameter of ~20 μm, is subject to strong drift even at distances of 10 cm or more from the firing surface. It should also be noted that the experiments were conducted in the laboratory under complete windless conditions. It is therefore easy to assume that even under sheltered growing conditions, i.e., when the air masses are relatively stable, as may occur in greenhouse conditions, the problem of drift is still considerable. The problem of very fine droplets is well known in the literature [31], and although they may offer advantages, such as better coverage, reduced drip losses, and possibly stomata penetration, these are undoubtedly negated by the effects of drift. In this context, the use of ST20 with varying potential did not provide any benefits. The ST50 system showed high coverage efficiency in the 5-20 cm range tested. Consequently, subsequent tests with both herbicides and beneficial microorganisms have shown that this system can be used for growing microgreens. In addition, from an application standpoint, the integration of the ink-jet system into DEMO was successful and could be adapted to microgreens growing systems with small plants and heights less than 15-20 cm and limited leaf area. However, extending it to other cropping systems is a major challenge. First of all, in its current configuration, the system can only apply very small amounts of liquid. Currently, a gradual reduction in the amounts used for pesticide treatment is observed in agricultural production. Specifically, the amounts used range from 200-500 l/ha to over 1500 l/ha. The ST50 system is currently not capable of even approaching the lower end of this spectrum. With this in mind, developing a system that uses ST50 in parallel could be an opportunity. Another option to be explored could be the development of a delivery system capable of producing larger droplets in the spectrum between 100 and 200 μm. This system could be capable of delivering larger volumes and therefore could be compared to systems currently used for pesticide distribution (e.g., mechanical and pneumatic sprayers) in other cropping systems (protected and field crops). In conclusion, this study, for the first time, demonstrated a potential application of the ink-jet technology for the distribution of agrochemical. Future research will be needed to evaluate the real applicability of this technology to different cultivation systems."
Round 2
Reviewer 3 Report
I accept the replies and modifications.